# Influence of Boundary Migration Induced Softening on the Steady State of Discontinuous Dynamic Recrystallization

**DOI:** 10.3390/ma14133531

**Published:** 2021-06-24

**Authors:** Frank Montheillet

**Affiliations:** CNRS, UMR 5307 Laboratoire Georges Friedel, Centre SMS, Mines Saint-Etienne, University of Lyon, F-42023 Saint-Etienne, France; montheil@emse.fr

**Keywords:** recrystallization (dynamic), grain boundary migration, recovery (dynamic), steady state, modeling

## Abstract

During discontinuous dynamic recrystallization (DDRX), new dislocation-free grains progressively replace the initially strain-hardened grains. Furthermore, the grain boundary migration associated with dislocation elimination partially opposes strain hardening, thus adding up to dynamic recovery. This effect, referred to as boundary migration induced softening (BMIS) is generally not accounted for by DDRX models, in particular by “mean-field” approaches. In this paper, BMIS is first defined and then analyzed in detail. The basic equations of a grain scale DDRX model, involving the classical Yoshie–Laasraoui–Jonas equation for strain hardening and dynamic recovery and including BMIS are described. A steady state condition equation is then used to derive the average dislocation density and the average grain size. It is then possible to assess the respective influences of BMIS and dynamic recovery on the strain rate sensitivity, the apparent activation energy, and the relationship between flow stress and average grain size (“Derby exponent”) of the material during steady state DDRX. Finally, the possible influence of BMIS on the estimation of grain boundary mobility and nucleation rate from experimental data is addressed.

## 1. Introduction

Dynamic recrystallization plays an essential role in hot forming of metals, since it gives them the ability to undergo significant deformations without cracking or fracturing due to the annealing of the major fraction of dislocations introduced by strain. Furthermore, dynamic recrystallization determines the main features of the deformed microstructures, mainly the grain size and the dislocation density, which in turn are reflected in the final mechanical properties, such as yield strength and ductility. It is thus very important to acquire a good knowledge of the involved metallurgical mechanisms to be able to control dynamic recrystallization and eventually to optimize industrial hot working processes. This point has already been highlighted a long time ago, for instance by Jonas and coworkers [1,2], and more recently by Sakai et al. [3].

The mechanism of discontinuous dynamic recrystallization (DDRX), which occurs in low to medium stacking fault energy materials, like copper, austenitic steels, and Ni or nickel alloys, is at first glance quite similar to static recrystallization (SRX): new dislocation-free grains nucleate and grow at the expense of the “old” strain-hardened grains. There are, however, significant differences:
-In SRX, the main softening effect is due to the migration of grain boundaries of the (recrystallized) growing grains, towards regions containing high dislocation densities: it is generally considered that dislocations are completely or almost completely annihilated by their interactions with moving grain boundaries. This mechanism can be referred to as boundary migration induced softening (BMIS). Static recovery within the regions not yet recrystallized constitutes a secondary softening process during SRX. In this way, the material transforms from an initial state of high dislocation density $\rho_{ini}$ due to prior deformation to a final state with very low dislocation density $\rho_{0}\approx \;0$. This makes SRX very similar to a phase change.-In DDRX, a more complex situation arises, because the recrystallized grains undergo strain-hardening during their growth. There are then three mechanisms that counterbalance the increase in dislocation density: (1) Dynamic recovery, which takes place in growing as well as shrinking grains; (2) The substitution of “younger” grains for “older” grains with higher dislocation content, whatever the interaction between dislocations and moving boundaries, which is merely a geometric effect, and (3) The annihilation of dislocations by grain boundary migration (BMIS), which is a physical effect. Note that mechanisms (2) and (3) are closely interrelated, although they can be clearly distinguished in theory. At large strain (von Mises equivalent strains of the order of 1), DDRX leads to a steady state where the material behaves as a dissipative structure that converts the mechanical energy input into heat.

A number of DDRX models have been proposed since the pioneering works of the 1970s [4,5] (see reviews [6,7,8]). They can be roughly classified into two groups: In the mesoscale, or grain-scale approaches (sometimes referred to as mean field models), the material is generally considered as a set of interrelating spherical grains, each of them being characterized by its diameter and dislocation density. Such models lead to analytical or quasi-analytical results and make it possible to investigate the steady state behaviour at large strains [9,10], possibly even including topological effects [11]. In the full field approaches, a complete description of the microstructure is considered, including for inon the basstance local dislocation densities, grain boundary mobilities and misorientations. These models initially involved Monte Carlo calculations [12,13,14] or cellular automata (e.g., [15,16,17]). Some authors later combined a cellular automata simulation of recrystallization with a finite element computation of deformation [18]. Recent full field models are based on a level set description of the grain boundaries within a finite element framework [19]. However, reaching large strains remains a computational challenge.

It is worthwhile to note that in all the above models BMIS is not taken into account, even not explicitly mentioned, and its potential effects on flow stress and average grain size are not discussed. As a result, a bias may be introduced when experimental data are analyzed on the basis of such models. The aim of this paper is to consider the issue in the simple case of the steady state by the extension of a previously published model [10,20]. The involved calculations are mainly analytical and closed form results are exhibited. In the next section, the basic equations of the model are shown, with a special emphasis on the incorporation of BMIS. The expressions of the steady state flow stress and average grain size are then derived in Section 3 and preliminary results are shown using data pertaining to a Ni-1% Nb alloy. In Section 4, relationships between the microscopic and macroscopic constitutive parameters of the material are established, and some numerical examples are given. Finally, a first set of conclusions is given about the influence of BMIS during steady state DDRX, and future developments are suggested.

## 2. Basic Equations

### 2.1. General Formulation of BMIS

First consider the case of a grain growing within a polycrystal, with volume *V* and average dislocation density *ρ* at time *t* (Figure 1). The length of dislocations contained in the grain is thus L= ρV. During an increment of time dt, this yields (Equation (1)):(1)$$d\rho =\;\rho \left(\frac{dL}{L}-\;\frac{dV}{V}\right)$$
where dL is the sum of two components (Equation (2)):(2)$$dL=\;d\rho_{h}V+\;(1-\;\delta )\rho dV$$
where, $d\rho_{h}$ is the increase in dislocation density due to strain hardening in volume *V* and dV is the increment of volume of the grain, while *δ* specifies the effect of the softening due to grain boundary migration. If δ= 0, the dislocation density in the volume dV swept by the boundary remains equal to *ρ*, which means that no BMIS occurs. By contrast if δ= 1, all dislocations present in the volume dV are absorbed by the moving boundary. Intermediate values of *δ* can also be considered to account for a possible partial BMIS effect, although there is no clear experimental evidence of such a case. Substituting Equation (2) into (1) and dividing by the strain increment dε during time dt leads to (Equation (3)):(3)$$\frac{d\rho }{d\varepsilon }=\;\frac{d\rho_{h}}{d\varepsilon }-\;\delta \frac{\rho }{V}\frac{dV}{d\varepsilon }$$
which is a generalized strain hardening equation including BMIS.

The case of a shrinking grain is different, since the volume dV swept by the boundary is removed, instead of appended, to the grain, such that the softening effect is absent.

### 2.2. Grain Growth and Strain Hardening Equations

A mesoscale approach is used here, where each grain of the aggregate is considered as a spherical inclusion of diameter *D* embedded in a uniform matrix of current dislocation density $\bar{\rho }$. It has been shown elsewhere that volume conservation implies to define $\bar{\rho }$ as the average dislocation density of all the grains *i*, weighted by the square of their diameter: $\bar{\rho }=\;{\sum_{i}\rho_{i}D_{i}}^{2}/{\sum_{i}D_{i}}^{2}$ [9,10]. Then, for a grain of diameter *D* and dislocation density *ρ*, the (algebraic) growth equation can be written (Equation (4)):(4)$$\frac{dD}{d\varepsilon }=\;\frac{2M\tau }{\dot{\varepsilon }}(\bar{\rho }-\;\rho )$$
where the strain rate $\dot{\varepsilon }$ is assumed identical for all the grains, *M* is the mobility of the grain boundaries, and *τ* the line energy of dislocations. This equation shows that when $\rho <\;\bar{\rho }$ the grain grows and conversely, when $\rho >\;\bar{\rho }$ the grain shrinks. Note that the product $\tau (\bar{\rho }-\;\rho )$ is the driving force for the migration of the grain boundary.

For strain hardening, Equation (3) must be specified. Various formulations have been proposed for the first term $d\rho_{h}/d\varepsilon$, e.g., the power law, the Kocks–Mecking, and Yoshie–Laasraoui–Jonas equations. It has been shown that each of them can be fitted fairly well with the same set of stress–strain data [21]. The Yoshie–Laasraoui–Jonas (YLJ) equation is well adapted for large strain hot deformations involved in metallurgical processing and it will therefore be used here. For a spherical grain of diameter *D*, dV/V= 3dD/D, Equation (3) can then be written (Equation (5)):(5)$$\frac{d\rho }{d\varepsilon }=\;h-\;r\rho -\;3\delta \frac{\rho }{D}\frac{dD}{d\varepsilon }$$
where *h* and *r* are two material parameters associated with strain hardening and dynamic recovery, respectively. Note that *h* has dimension of a dislocation density, while *r* has no dimension. The (non-dimensional) BMIS parameter *δ* ranges from 0 (no BMIS) to 1 (full BMIS) when the grain grows and is zero when the grain shrinks.

A third equation is needed for modeling nucleation, which is a key factor in the DDRX process. However, as far as the steady state is concerned, the exact time and the specific physical mechanisms of nucleation (e.g., grain boundary bulging, growth of strongly misoriented subgrains, twinning, …) are not of prime importance: the main point is to state that each grain gives birth, on average, to one single new grain in its lifetime [9,10]. This is contained in Equation (11) below.

### 2.3. Introduction of Non-Dimensional Variables

In former works, the dislocation density was normalized by $\rho_{\infty }=\;h/r$ which is the asymptotic value of *ρ* for the YLJ equation, i.e., when only strain hardening and dynamic recovery are operating. However, this choice causes difficulties to deal with the case where dynamic recovery is very low, since then $\rho_{\infty }\to \;\infty$. The non-dimensional variable z= ρ/h is thus introduced, which takes value $\xi =\;\bar{\rho }/h$ for the average dislocation density. Since due to DDRX $\bar{\rho }<\;\rho_{\infty }=\;h/r$, ξ< 1/r. Equation (4) can then be written (Equation (6)):(6)$$\frac{dD}{d\varepsilon }=\;\frac{2M\tau h}{\dot{\varepsilon }}(\xi -\;z)=\;D^{*}(\xi -\;z)$$
where $D^{*}$ is a reference grain size. Then defining $y=\;D/D^{*}$ (Equation (7)):(7)$$\frac{dy}{d\varepsilon }=\;\xi -\;z$$

Equation (5) can then be re-written in the non-dimensional form (Equation (8)):(8)$$\frac{dz}{d\varepsilon }=\;1-\;rz-\;3\delta \frac{z}{y}\frac{dy}{d\varepsilon }$$

Finally, combining Equations (7) and (8) gives (Equation (9)):(9)$$\frac{dz}{d\varepsilon }=\;1-\;rz-\;3\delta \frac{z(\xi -\;z)}{y}$$

Numerical resolution of the two coupled differential Equations (7) and (9), for the two functions *y* (grain size) and *z* (dislocation density) will give the evolution of a given grain of the system. In steady state, *ξ* remains constant during the grain life. The two functions y(ε) and z(ε) depend only on the dynamic recovery *r* and BMIS *δ* parameters. If the latter are assumed to be identical for all the grains, as well as the hardening parameter *h* and the grain boundary mobility *M*, all grains will follow the same evolution as a function of their specific strain *ε*.

As initial conditions, it will be assumed that a grain nucleates with zero diameter and dislocation density: for ε= 0, y= z= 0. According to Equation (7), the initial slope of *y* is *ξ*. Substituting this value into Equation (9) for ε= 0 leads to (Equation (10)):(10)$$\frac{dz}{d\varepsilon }=\;\frac{1}{1+\;3\delta }$$

This merely means that, as expected, the initial increase in the dislocation density is moderated by BMIS.

For z= ξ (i.e., $\rho =\;\bar{\rho }$), the grain size goes through a maximum corresponding to a strain $\varepsilon_{M}$, and for some strain ε= ω, *y* goes to zero which means that the grain disappears.

## 3. Derivation of the Steady State Flow Stress and Grain Size

### 3.1. Steady State Condition

It was mentioned above that upon steady state, each grain gives birth to one and only one new grain in its lifetime. This condition can be written in the form (Equation (11)):(11)$$\frac{k_{N}{\bar{\rho }}^{p}}{\dot{\varepsilon }}\int_{0}^{\omega }D^{2}d\varepsilon =\;1$$
where the integral is extended over the strain interval [0, *ω*]. The squared grain size *D* means that DDRX nucleation is known to occur mainly at grain boundaries, whereas a nucleation in volume would imply the factor $D^{3}$. $k_{N}$ and *p* are two material parameters characterizing the nucleation rate. The exponent *p* can be set to 3, as shown in previous work, which reflects the fact that the nucleation rate increases rapidly with the overall dislocation density of the material.

Substituting *y* for *D* in the above equation yields (Equation (12a–c)):(12a)$$\xi^{3}G(\xi ,r)=A$$
in which *A* is a constant for given strain rate, temperature and material, *viz.*:(12b)$$A=\;\frac{{\dot{\varepsilon }}^{3}}{4\;k_{N}M^{2}\tau^{2}h^{5}}$$

And G(ξ,r) denotes the (non-analytical) integral:(12c)$$G(\xi ,r)=\;\int_{0}^{\omega }y^{2}d\varepsilon$$

The product $\xi^{3}G(\xi ,r)$ is plotted in Figure 2 as a function of *ξ* for various values of *δ* and *r* in a double logarithmic reference frame. It appears clearly that the data are very well fitted by straight lines, which means that $\xi^{3}G(\xi ,r)$ is a power law function of *ξ* (this would of course be true as well for any other nucleation exponent *p*). Equation (12a) can then be written in the form (Equation (13)):(13)$$k_{1}\;\xi^{q_{1}}=\;A$$
which allows $\xi =\;\bar{\rho }/h$ to be readily calculated. Table 1 gives the values of the non-dimensional parameters $k_{1}$ and $q_{1}$ deduced by linear regressions from the curves of Figure 2. For given *r*, $k_{1}$ increases with δ, while the slope $q_{1}$ is mainly dependent of *r*.

The steady state flow stress is finally derived from the classical Taylor equation $\sigma_{s}=\;\alpha \mu \;b\sqrt{\bar{\rho }}=\;\alpha \mu \;b\sqrt{h\xi }$, which yields (Equation (14)):(14)$$\frac{\sigma_{s}}{\alpha \mu b}=\;\sqrt{h}{\left(\frac{{\dot{\varepsilon }}^{3}}{4k_{N}M^{2}\tau^{2}h^{5}k_{1}}\right)}^{1/2q_{1}}$$

To deal with a numerical example, data pertaining to a model Nickel-1% Niobium alloy deformed at 900 °C and 0.01 $s^{-1}$ will be used [20,22]: $h=\;1000\;\mu m^{-2},$ $k_{N}=\;5\times 10^{-9}\;\mu m^{4}s^{-1}$ (with p= 3), $M\tau =\;0.1\;\mu m^{3}s^{-1}$. Measurements have shown that r= 5, but *r* will be varied here between 0 and 10 to assess the influence of dynamic recovery, a range that is typical for materials undergoing DDRX. Results are displayed in Figure 3, which shows that BMIS is significantly larger (i.e., when *δ* is increased from 0 to 1) than softening induced by dynamic recovery (i.e., when *r* is increased from 0 to 10).

The same approach is followed now for the steady state grain size.

### 3.2. Average Steady State Grain Size

The average grain size $D_{s}$ during steady state is defined here, as it is most often the case, by the number-weighted grain diameter, which can be expressed by the integral (Equation (15)):(15)$$D_{s}=\;\frac{1}{\omega }\int_{0}^{\omega }D\;d\varepsilon =\;\frac{D^{*}}{\omega }\int_{0}^{\omega }y\;d\varepsilon$$

In this equation, the integral as well as the lifetime of the grain *ω* depend only on *ξ* and *r*, and Equation (15) can be written (Equation (16)):(16)$$\frac{D_{s}}{D^{*}}=\;\frac{H(\xi ,r)}{\omega (\xi ,r)}$$

The ratio H(ξ,r)/ω(ξ,r) is plotted in Figure 4 with respect to *ξ* for various values of *δ* and *r* in a double logarithmic set of axes. The data are very well aligned, such that Equation (16) can be written (Equation (17)):(17)$$\frac{D_{s}}{D^{*}}=\;k_{2}\xi^{q_{2}}$$
where the values of the non-dimensional parameters $k_{2}$ and $q_{2}$ are reported in Table 2. It is important to remember that these two parameters, as well as their counterparts $k_{1}$ and $q_{1}$ depend only on *r* and *δ*, which makes the above results applicable to any material. It appears that $k_{2}$ increases with *δ* and, to a lesser extent, with *r*, while the exponent $q_{2}$ increases with *r* but is almost independent of *δ*.

Figure 5 shows, with the same data as above, that $D_{s}$ is significantly increased by BMIS and much less by dynamic recovery, which is qualitatively consistent with the evolutions of $\sigma_{s}$ depicted in Figure 3.

### 3.3. Grain Size and Dislocation Density Changes Along the Lifetime of a Grain

Numerical resolution of Equations (7) and (9) provides access to the two functions y(ε) and z(ε) which are associated with the grain size $D=\;D^{*}y$ and the dislocation density ρ= hz, respectively. The strain dependence of *D* and *ρ* for a grain of the aggregate is illustrated in Figure 6a,b for various values of *δ* and *r*. The same material parameters as above were used, *ξ* being determined from Equation (13). It is important to remind here that such curves are identical for all grains in steady state since their constitutive parameters are assumed to be the same and $\bar{\rho }$ is uniform throughout the material.

Figure 6a shows the influence of BMIS in the absence of dynamic recovery. The case δ= 0 is special because Equations (7) and (9) can be solved analytically [9,10]: *ρ* increases linearly and the curve D(ε) is merely a parabola, which is not the case otherwise. For a given strain, BMIS increases the grain size and the ultimate strain *ω*, but decreases the dislocation density. Figure 6b exhibits in turn the influence of dynamic recovery when full BMIS occurs (δ= 1). Dynamic recovery is mainly sensitive in the second half of the grain lifetime, where it increases the grain size and decreases the dislocation density. The maximum grain size, however, remains almost unchanged by dynamic recovery. In addition, BMIS and recovery both increase the lifetime *ω* which ranges from 0.091 for δ= r= 0 to 0.165 for δ= 1 and r= 10.

Another parameter allowing to assess the evolution of a grain is the aspect ratio *λ*. In uniaxial compression, the latter is defined as the ratio between the axis parallel to the compression direction and a perpendicular axis, which is given by (Equation (18)):(18)$$\lambda =\;exp(-\;\frac{3}{2}\varepsilon )$$
where *ε* denotes here the von Mises equivalent strain. In simple shear (corresponding to the classical torsion test), the relationship between *λ* and *ε* is more complex, but it is easy to show that it is numerically very close to Equation (18) (see for instance [23]).

Figure 7 shows the dependence of *λ* on BMIS for various values of *r* in two cases: (i) for $\varepsilon =\;\varepsilon_{M}$, i.e., when the grain reaches its maximum size, and (ii) for ε= ω, i.e., when it vanishes. This second case may appear less relevant since the grain size then approaches zero, but it gives the minimum aspect ratio value for given deformation conditions. It can be seen that the two softening effects contribute to a moderate decrease in *λ* in similar proportions. The two sketches on the right side of the diagram illustrate the shape of the grain for δ= 1 and r= 10 by a section through its revolution axis. It appears that such a grain can still be termed “equiaxial” in the traditional sense of the word. This observation justifies the initial assumption of spherical grains in the model.

## 4. Constitutive Parameters

The aim of this section is to derive the macroscopic constitutive parameters of the material during DDRX steady state from the microscopic parameters associated with the elementary mechanisms of strain hardening (*h*) and dynamic recovery (*r*), nucleation ($k_{N}$ and *p*) and grain growth (Mτ).

### 4.1. Microscopic Constitutive Parameters

Stress–strain curves at various strain rates and temperatures allow *h* and *r* to be readily derived, and it has been observed that r≈ 5 remains approximately constant in low stacking fault energy materials like austenitic stainless steels or Ni-Nb alloys [22], while *h* takes the form (Equation (19)):(19)$$h=\;h_{0}({\dot{\varepsilon }/{\dot{\varepsilon }}_{0})}^{m_{h}}exp(m_{h}Q_{h}/RT)$$

Here, $m_{h}$ and $Q_{h}$ are the strain rate sensitivity and apparent activation energy specific to strain hardening, ${\dot{\varepsilon }}_{0}$ is a reference strain rate (we take in the following ${\dot{\varepsilon }}_{0}=\;1\;s^{-1}$), $h_{0}$ is a material constant and *R* is the gas constant.

The grain boundary mobility *M* and nucleation rate $k_{N}$ parameters can be determined in turn from the measured steady state flow stress $\sigma_{s}$ and average steady state grain size $D_{s}$ (see Section 4.4 below) [10,24]. It was shown that they can be written (Equations (20) and (21)):(20)$$M=\;M_{0}({\dot{\varepsilon }/{\dot{\varepsilon }}_{0})}^{m_{M}}exp(-\;Q_{M}/RT)$$
(21)$$k_{N}=\;k_{N_{0}}({\dot{\varepsilon }/{\dot{\varepsilon }}_{0})}^{m_{N}}exp(-\;Q_{N}/RT)$$
where $m_{M}$ and $m_{N}$, $Q_{M}$ and $Q_{N}$ are the rate sensitivities and activation energies associated with migration and nucleation, respectively, and $M_{0}$ and $k_{N_{0}}$ are constants. It is recalled here that the nucleation exponent *p* is set at 3. Furthermore, parameters $k_{1}$, $q_{1}$ and $k_{2}$, $q_{2}$ in Table 1 and Table 2 depend only on *r* and *δ* and are thus independent of strain rate and temperature.

### 4.2. Macroscopic Strain Rate Sensitivity and Activation Energy

The strain rate sensitivity $m_{DRX}=\;({\partial \ln\sigma_{s}/\partial \ln\dot{\varepsilon })}_{T}$ (at a given temperature) is derived from Equation (14), which gives (Equation (22a)):(22a)$$m_{DRX}=\;\frac{m_{h}}{2}+\;\frac{1}{2q_{1}}\left(3-\;m_{N}-\;2m_{M}-\;5m_{h}\right)$$

In a similar way, the apparent activation energy $Q_{DRX}=\;(R/m_{DRX}){\left[\partial \ln\sigma_{s}/\partial (1/T)\right]}_{\dot{\varepsilon }}$ (at a given strain rate) is derived from (14) to get (Equation (22b)):(22b)$$Q_{DRX}=\;\frac{m_{h}Q_{h}+\;(1/q_{1})(Q_{N}+\;2Q_{M}-\;5m_{h}Q_{h})}{m_{h}+\;(1/q_{1})(3-\;m_{N}-\;2m_{M}-\;5m_{h})}$$

From the above formulae, the influence on $m_{DRX}$ and $Q_{DRX}$ of each of the microscopic parameters taken separately can be assessed for any given material. Nevertheless, only the simple case where $m_{N}=\;m_{M}=\;m_{h}=\;m$ and $Q_{N}=\;Q_{M}=\;Q_{h}=\;Q$ will be examined further here. Equation (22a,b) then reduce to (Equation (23a,b)):(23a)$$m_{DRX}=\;\frac{1}{2}\left(1-\;\frac{8}{q_{1}}\right)m+\;\frac{3}{2q_{1}}$$
(23b)$$Q_{DRX}=\;\frac{\left(q_{1}-\;5\right)m+\;3}{\left(q_{1}-\;8\right)m+\;3}Q$$

As shown in Figure 8, $m_{DRX}$ is a weakly decreasing (for r= 0) or increasing (for r= 5 or 10) linear function of *m*. However, it does not depend on BMIS significantly. An important point is that $m_{DRX}$ does not vanish when m= 0, which means that the steady state strain rate sensitivity is mainly an intrinsic effect of DDRX and does not simply reflect the microscopic rate sensitivities. $Q_{DRX}$ is in turn almost insensitive to *r* as well as to *δ*, as can be checked numerically. Equation (23b) shows that it is always slightly larger than its microscopic counterpart *Q*. More specifically, since $q_{1}\approx \;8$, $Q_{DRX}=\;(1+\;m)\;Q$ to a first approximation. Finally, $Q_{DRX}$ vanishes with *Q* which indicates that the apparent activation energy for DDRX steady state arises directly from the underlying microscopic mechanisms.

### 4.3. Derby Exponent

It has long been observed experimentally that the steady state flow stress $\sigma_{s}$ and average grain size $D_{s}$ are related by an inverse power law equation $\sigma_{s}=\;K/D_{s}{}^{a}$ where *K* is roughly independent of temperature and strain rate for a given material, and the exponent *a* ranges between 0.5 and 1 [25,26,27]. Accordingly, this result has also been obtained from the mesoscopic model used in the present paper [10], and this *Derby equation* can be considered as quite specific to the steady state of discontinuous dynamic recrystallization. It is thus interesting to assess the possible influence of BMIS on the exponent *a*.

By eliminating $\dot{\varepsilon }$ between the two Equations (14) and (17), the following inverse power relationship is obtained (Equation (24)):(24)$$\sigma_{s}=\;\alpha \mu b\sqrt{h}{\left[\frac{2M\tau }{k_{N}h^{2}}\frac{k_{2}{}^{3}}{k_{1}}\right]}^{\frac{1}{2(q_{1}-3q_{2})}}\frac{1}{D_{s}{}^{\frac{3}{2(q_{1}-3q_{2})}}}$$
which makes appear the exponent (Equation (25)):(25)$$a=\;\frac{3}{2(q_{1}-\;3q_{2})}$$

If the factor before the inverse power of $D_{s}$ is independent of temperature, the Derby equation is verified. In addition, since $q_{1}\approx \;8$ and $q_{2}\approx \;2$ (see Table 1 and Table 2), a≈ 0.75 as expected, and it depends only on *r* (dynamic recovery) and *δ* (BMIS). Figure 9 shows the influence of *δ* on the exponent *a* for various values of *r*. It is not easy to draw conclusions from this diagram, since the curve for r= 10 is not monotonic. It appears nevertheless that a≈ 0.75 whatever *r*, when BMIS is taken into account, while it decreases with increasing dynamic recovery in the absence of BMIS. Since BMIS actually occurs during hot deformation, this is in line with the experimental observations.

### 4.4. Estimation of Mτ and $k_{N}$ from the Experimental Data

In all the numerical examples presented above, the boundary migration and nucleation parameters Mτ and $k_{N}$ were supposed to be known. Although direct measurements are impossible during DDRX steady state, it is indeed possible to solve Equations (14) and (17) for these two quantities, assuming that the other measurable microscopic parameters have been already determined. However, this has been done so far without taking BMIS into account [10]. In this last subsection, the influence of BMIS and the recovery parameter *r* on the estimation of Mτ and $k_{N}$ is analyzed.

As the starting point, parameters *h*, *r*, $\sigma_{s}$ and $D_{s}$ are supposed to be known, as well as the shear modulus μ and the Burgers vector *b* at the investigation temperature (and, of course, the prescribed strain rate $\dot{\varepsilon }$). The reduced steady state dislocation density ξ is first derived from the Taylor equation $\sigma_{s}=\;\alpha \mu \;b\sqrt{h\xi }$. Equations (14) and (17) can then be solved for Mτ and $k_{N}$, which yields (Equation (26a,b)):(26a)$$M\tau =\;\frac{\dot{\varepsilon }D_{s}}{2\;hk_{2}\xi^{q_{2}}}$$
(26b)$$k_{N}=\;\frac{\dot{\varepsilon }k_{2}{}^{2}}{h^{3}D_{s}{}^{2}k_{1}\xi^{q_{1}-2q_{2}}}$$

Values of Mτ and $k_{N}$ obtained from the above equations are reported in Table 3 for three levels of BMIS and three values of the dynamic recovery parameter *r*. For each line of the table, the values of ξ and $D_{s}$ derived from the $k_{1}$, $k_{2}$, $q_{1}$, $q_{2}$ parameters in column δ=1 in Table 1 and Table 2 were used. The values $M\tau =\;0.1\;\mu m^{3}s^{-1}$ and $k_{N}=\;5\times 10^{-9}\;\mu m^{4}s^{-1}$ introduced in Section 3 are thus logically recovered for δ=1. For δ=0.5 and δ=0 (i.e., in the absence of BMIS), larger values of Mτ and $k_{N}$ are observed, whereas the influence of *r* is weak.

This means that, for the same set of experimental data, neglecting BMIS leads to a significant overestimation of grain boundary mobility and nucleation rate. This is easy to understand since BMIS favors both flow softening and average grain size increase, in the same manner as an increase in mobility or nucleation rate.

## 5. Conclusions and Future Developments

In this paper, a mesoscale model of discontinuous dynamic recrystallization (DDRX) was used to investigate the various effects of boundary migration induced softening (BMIS) during the steady state. BMIS is a mechanism specific to DDRX, which acts in addition to dynamic recovery and the nucleation and growth of new grains, by removing dislocations from the volumes swept by the moving grain boundaries. Dynamic recovery effects were also taken into consideration and compared to the latter. The main outcomes are the following:(i).As expected, BMIS induces significant flow softening. For *r* values ranging between 0 and 10, typical of materials undergoing DDRX, BMIS is even more efficient than dynamic recovery.(ii).The second major effect of BMIS is to promote average grain size growth, while the influence of dynamic recovery is weak.(iii).The lifetime of a grain, and thus the strain at the time of disappearance, is increased by BMIS. The aspect ratio of the grains nevertheless remains sufficiently close to unity for them to be considered as approximately equiaxed.(iv).By contrast, the macroscopic strain rate sensitivity $m_{DRX}$ and apparent activation energy $Q_{DRX}$ are not considerably modified by BMIS.(v).The classical Derby equation relating the flow stress to the average grain size was found by the model. The Derby exponent *a* is globally increased by BMIS. For full BMIS (δ= 1), it takes a value close to 0.75 whatever the level of dynamic recovery.(vi).Finally, the present approach shows that whenever the grain boundary migration parameter Mτ and the nucleation rate parameter $k_{N}$ are estimated from the data with the mesoscale model, they can be both overestimated if BMIS is neglected.

In the future, insofar as complete sets of microscopic constitutive parameters for a given material at various strain rates and temperatures are available, further tests should be carried out to assess the validity of the model and confirm the present results. Another interesting development would be to investigate the transient behaviour of the material, which occurs at low to moderate strains (e.g., ε< 0.8 − 1.0 in austenitic steels or copper [28,29,30]). In this case, in contrast to the steady state behaviour, the histories of the grains [i.e., the functions D(ε) and ρ(ε)] are all different from each other. As a result, average quantities over the grains cannot be replaced any longer by integrals over strain for one single grain, like in Equations (11), (12c), and (15). It is then necessary to consider a whole set of interacting grains and to compute their evolutions by applying the basic differential equations of the model to each of them. This has already been carried out [9,10]. In particular, it was shown that the model was able to predict the transition from multiple peak (low strain rate, high temperature) to single peak (high strain rate, low temperature) stress–strain curves. However, the specific effects of BMIS still remain to be investigated in this context.

Finally, the model could also be extended to include the grain shape changes under combined effects of migration and prescribed strain rate. A first step has already been taken in this direction, for the growth or dissolution of a particle in a matrix submitted to axisymmetric compression or simple shear, although again in the absence of BMIS [31].

## Figures and Tables

**Figure 1 materials-14-03531-f001:**
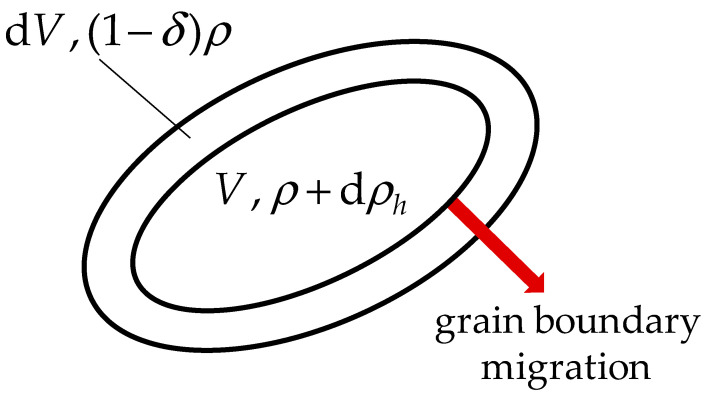
Schematic representation of a growing grain.

**Figure 2 materials-14-03531-f002:**
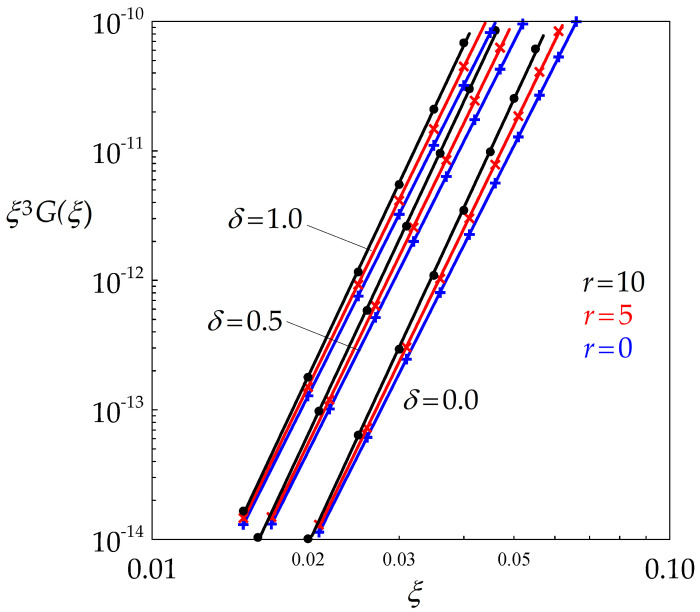
Double logarithmic diagram highlighting the power law dependence of $\xi^{3}G(\xi )$ with respect to ξ for various values of *r* and δ.

**Figure 3 materials-14-03531-f003:**
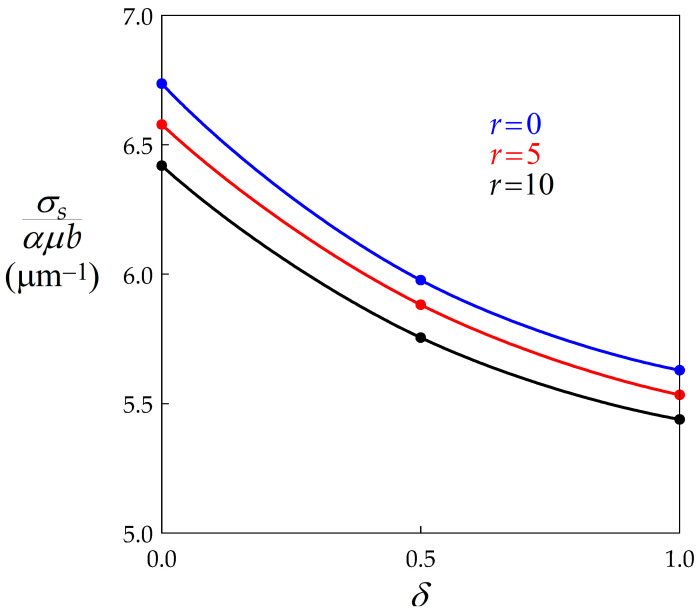
Diagram showing the dependence of the reduced steady state flow stress on the BMIS parameter for three values of the recovery coefficient *r*. Data points are connected by spline curves.

**Figure 4 materials-14-03531-f004:**
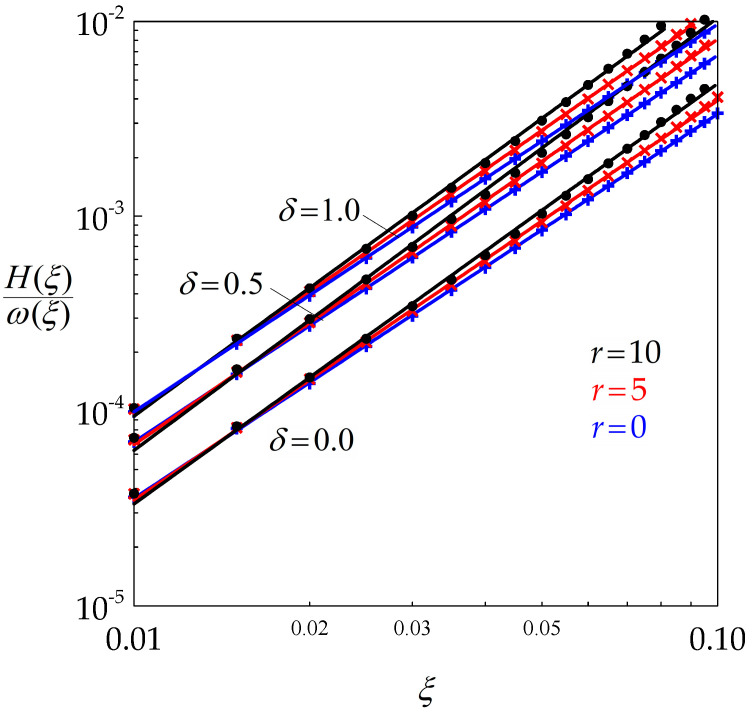
Double logarithmic diagram highlighting the power law dependence of H(ξ)/ω(ξ) with respect to ξ for various values of *r* and δ.

**Figure 5 materials-14-03531-f005:**
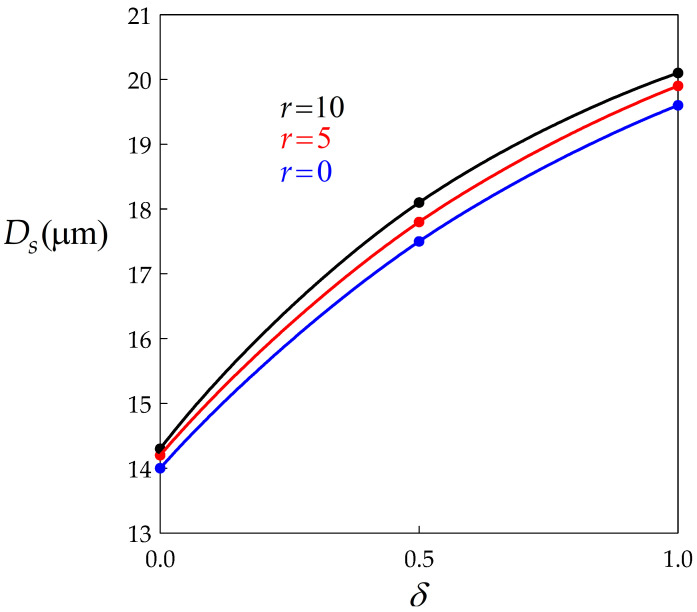
Diagram showing the dependence of the steady state grain size on the BMIS parameter for three values of the recovery coefficient *r*. Data points are connected by spline curves.

**Figure 6 materials-14-03531-f006:**
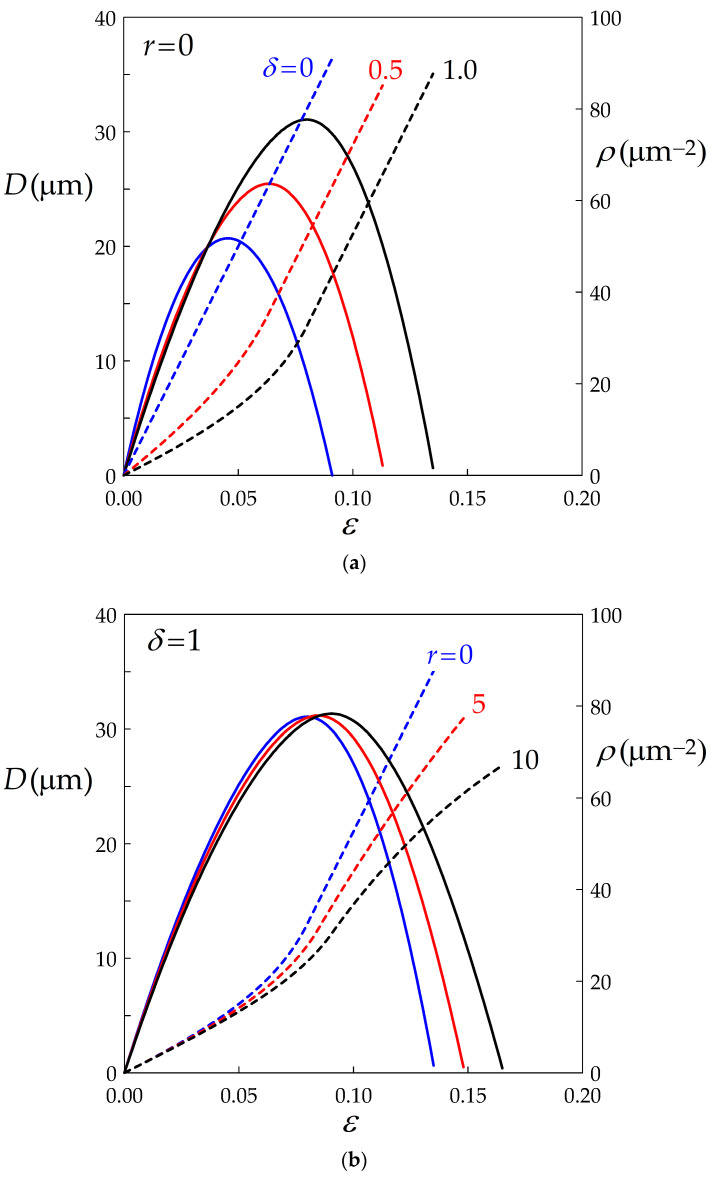
Strain dependence of the grain size (solid lines) and dislocation density (broken lines) for any grain in the aggregate; (**a**) No dynamic recovery: influence of BMIS; (**b**) Full BMIS: influence of dynamic recovery.

**Figure 7 materials-14-03531-f007:**
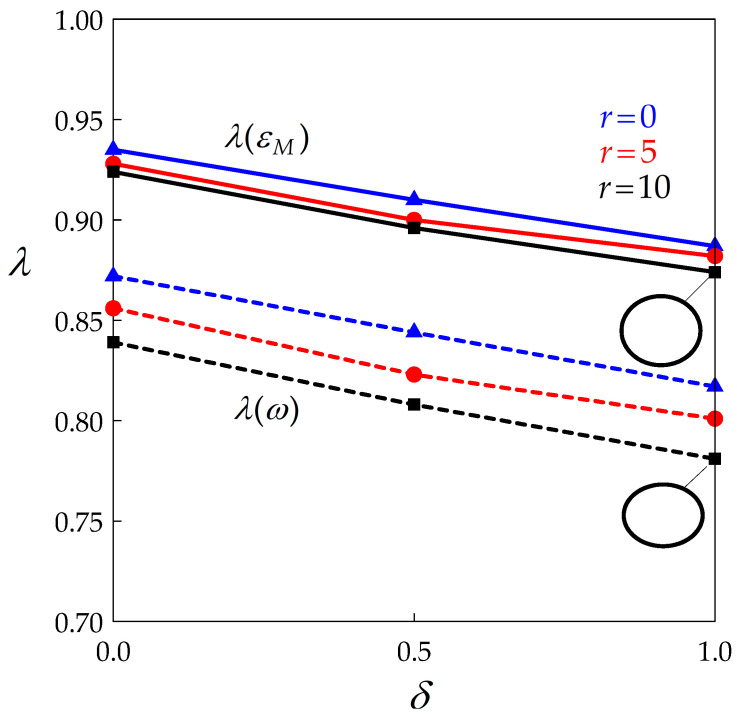
Dependence of the aspect ratio of a grain on BMIS at its maximum size ($\varepsilon =\;\varepsilon_{M}$) and when it vanishes (ε= ω) for three values of *r*. The two sketches on the right side of the diagram illustrate the shape of the grain.

**Figure 8 materials-14-03531-f008:**
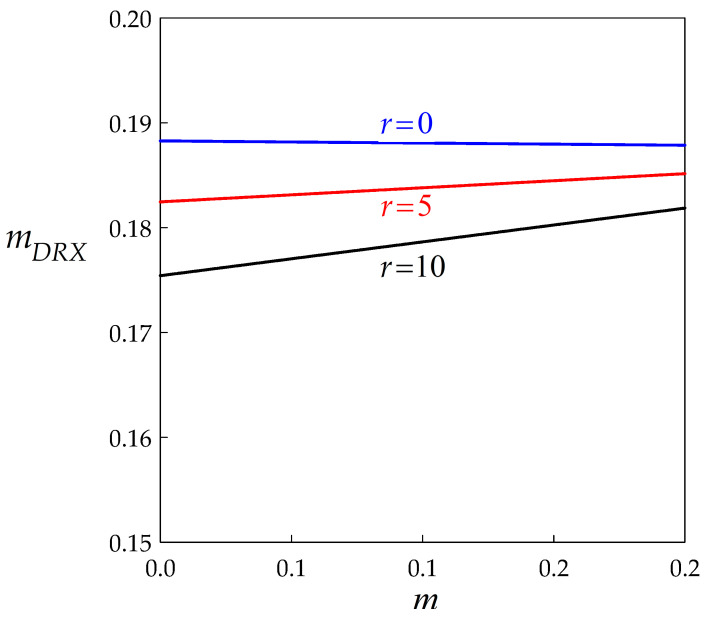
Influence of the microscopic strain rate sensitivity *m* on its macroscopic counterpart $m_{DRX}$ for three values of the dynamic recovery parameter.

**Figure 9 materials-14-03531-f009:**
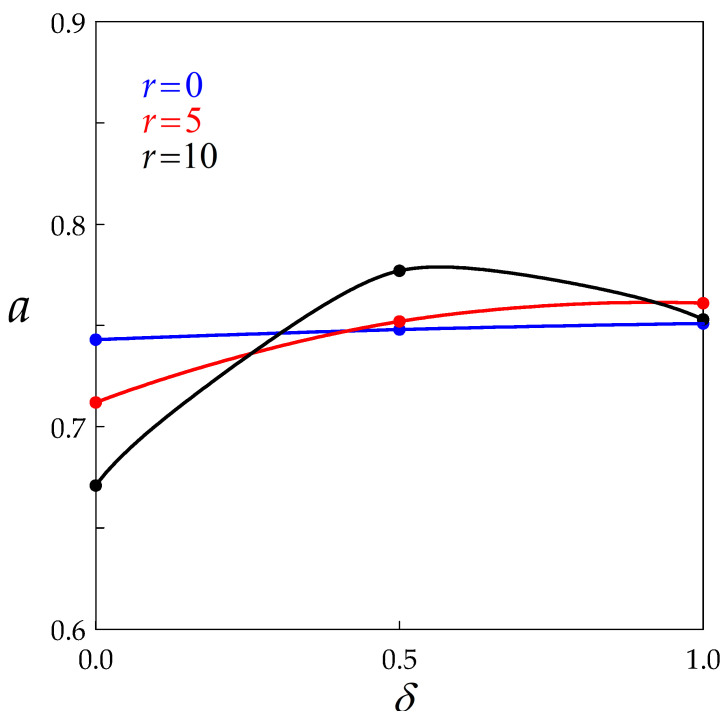
Dependence of the Derby exponent on BMIS for three values of the recovery parameter. Data points are connected by spline curves.

**Table 1 materials-14-03531-t001:** Numerical values of parameters $k_{1}$ and $q_{1}$ involved in Equation (13).

		BMIS Parameter *δ*					
		0		0.5		1.0	
		$k_{1}$	$q_{1}$	$k_{1}$	$q_{1}$	$k_{1}$	$q_{1}$
Dynamic recovery parameter *r*	0	0.233	7.937	1.587	7.960	4.399	7.967
	5	0.898	8.263	5.449	8.241	13.912	8.221
	10	5.551	9.709	25.144	8.591	60.200	8.551

**Table 2 materials-14-03531-t002:** Numerical values of parameters $k_{2}$ and $q_{2}$ involved in Equation (17).

		BMIS Parameter *δ*					
		0		0.5		1.0	
		$k_{2}$	$q_{2}$	$k_{2}$	$q_{2}$	$k_{2}$	$q_{2}$
Dynamic recovery parameter *r*	0	0.314	1.973	0.647	1.985	0.944	1.990
	5	0.443	2.052	0.976	2.082	1.416	2.083
	10	0.689	2.158	1.728	2.220	2.220	2.186

**Table 3 materials-14-03531-t003:** Numerical values of the migration parameter Mτ (μm^3^ s^−1^) and the nucleation parameter $k_{N}$ (10^−9^ μm^4^ s^−1^) derived from Equation (25a,b), respectively, for various values of the dynamic recovery and BMIS parameters.

		BMIS Parameter *δ*					
		0		0.5		1.0	
		Mτ	$k_{N}$	Mτ	$k_{N}$	Mτ	$k_{N}$
Dynamic recovery parameter *r*	0	0.278	10.18	0.140	6.33	0.100	5.00
	5	0.280	10.38	0.141	6.25	0.100	5.00
	10	0.282	10.37	0.140	6.16	0.100	5.00

## Data Availability

Data used in this paper are available in references [20,22].

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
