# Peer review of "Influence of Boundary Migration Induced Softening on the Steady State of Discontinuous Dynamic Recrystallization"

_materials, 2021, doi:10.3390/ma14133531_

Round 1

Reviewer 1 Report

The paper brings a nice model of discontinuous dynamic recrystallization using the boundary migration induced softening effect. The topic is relevant both for the Materials journal and the corresponding scientific community.

The paper is clearly presented and easy to read. The presented model scientically relevant and correct The prime novelty of this paper is the introduction od boundary migration induced softening effect to the model of dynamic recystallization. As far as I know, this approach is original and not published yet. The influence of this effect on the recrystallization process is described and discussed logically in line with the presentation of the results. The Conclusions are summarizing the main results well and bring the promise of interesting results coming in the near future. 

1.I could recommend minor improvements in the organization of the manuscript, e.g. there is a sentence “The data are very well aligned, such that Equation (16) can be written:”, leading to the equation, but there is a figure between the text and the formula.

2.There is quite a lot of self-cites. I recommend to add more references to the works of the other authors in this field.

3.The organization of the manusript does not follow the common structure (i.e. Introduction, Materials and Methods – Results – Discussion – Conclusions). However, it is quite common in the case of the papers focused on modeling. Maybe the separation of the used methodology, results and discussion would make the paper a bit more understandable for “conventional” readers.

Author Response

1.I could recommend minor improvements in the organization of the manuscript, e.g. there is a sentence “The data are very well aligned, such that Equation (16) can be written:”, leading to the equation, but there is a figure between the text and the formula.

2.There is quite a lot of self-cites. I recommend to add more references to the works of the other authors in this field.

3.The organization of the manusript does not follow the common structure (i.e. Introduction, Materials and Methods – Results – Discussion – Conclusions). However, it is quite common in the case of the papers focused on modeling. Maybe the separation of the used methodology, results and discussion would make the paper a bit more understandable for “conventional” readers.

Response to Referee #1.

The referee is acknowledged for his(her) recommendations. Figure 3 has been moved to restore the continuity of the text and a number of new references have been added.

The author is fully aware that the structure of the paper is different from that of experimental articles. However, the latter would not be relevant for the present modeling paper, where the appropriate schedule is rather Introduction - Basic equations (starting point) - Modeling developments and results - Conclusions, which is nevertheless understandable.

Reviewer 2 Report

  • First, I see very few specific for dynamic recristallization (DRX) process in proposed model. Author suggest that new grains appear randomly and hardening/recovery processes are develop during grain growth. However, the most important feature of DRX is nucleation new grains due to continued strain. This is not taken into account in  the considered model.  
  • Next, I don’t see also criteria when DRX start. Author considered shady state behaviour corresponding to rather large times. In the same time, there are an important features  (in particular, oscillating stress-strain behaviour) which develop in early stage.  This oscillating stress-strain behaviour is result of the alternation of processes of new grain nucleation and strain hardening. It seems difficult to take into account this feature  within proposed approach.  
  • Finally, I would like to pay author attention to the relative recent review: T. Sakai, A. Belyakov, R. Kaibyshev, H. Miura, J. J. Jonas, Dynamic and post-dynamic recrystallization under hot, cold and severe plastic deformation conditions Progress in Materials Science 60 (2014) 130–207.

Author Response

First, I see very few specific for dynamic recristallization (DRX) process in proposed model. Author suggest that new grains appear randomly and hardening/recovery processes are develop during grain growth. However, the most important feature of DRX is nucleation new grains due to continued strain. This is not taken into account in  the considered model. 

The referee is acknowledged for his(her) careful rereading of the manuscript and comments.

The main controversial issue is nucleation. The author agrees of course that it is a key parameter in the discontinuous dynamic recrystallization process. However, as far as the steady state is concerned, the exact time and the specific physical mechanisms of nucleation (e.g. grain boundary bulging, growth of strongly misoriented subgrains, twinning, …) are not of prime importance: the main point is to state that each grain gives birth, on average, to one single new grain along its lifetime. This is contained in Equation (11), which also accounts for a nucleation occurring at grain boundaries, with a rate increasing rapidly with the overall dislocation density of the material.

Next, I don’t see also criteria when DRX start. Author considered shady state behaviour corresponding to rather large times. In the same time, there are an important features  (in particular, oscillating stress-strain behaviour) which develop in early stage.  This oscillating stress-strain behaviour is result of the alternation of processes of new grain nucleation and strain hardening. It seems difficult to take into account this feature  within proposed approach.

The author recognizes that the paper is restricted to the analysis of the steady state, as mentioned in the title. The main reason is that this allows to derive simple results in closed form almost fully analytically. However, the steady state behaviour is not without practical relevance either, since it can be achieved for some metals and alloys (e.g. copper, nickel, austenitic steels, …) at moderate accumulated strains (< 1), which can be encountered in various industrial thermomechanical processes. The possibility of extending the present analysis to transient phenomena, like the occurrence of multiple peak (oscillating) stress-strain curves, has been mentioned in an additional paragraph of the conclusive section.

Finally, I would like to pay author attention to the relative recent review: T. Sakai, A. Belyakov, R. Kaibyshev, H. Miura, J. J. Jonas, Dynamic and post-dynamic recrystallization under hot, cold and severe plastic deformation conditions Progress in Materials Science 60 (2014) 130–207.

A number of new references, including the review paper of Sakai et al. (2014), have been introduced in the manuscript.

Reviewer 3 Report

The author presents a mean field model for discontinuous dynamic recrystallization that takes into account boundary migration induced softening (BMIS). This is clearly a unique, original contribution. The objective is clearly stated and put well into context. The model and its necessary equations are didactically detailed. This article is likely to be cited in the future and the reviewer will be looking forward to upcoming contributions.

Author Response

This article is likely to be cited in the future and the reviewer will be looking forward to upcoming contributions.

The referee is thanked for his(her) favourable consideration of the manuscript. Regarding possible future contributions in line with this paper, an additional paragraph has been appended to the conclusive section.

Round 2

Reviewer 2 Report

I am satisfied with the author's response to the comments of the first review. I believe that the presented work can be recommended for publication. Nevertheless, for more clarity, I would recommend the author to add his explanation to Remark 1 in the text of the article (section Grain Growth and Strain Hardening Equations)

Author Response

Response to Reviewer 2.

According to the referee’s recommendation, a short comment about nucleation has been appended at the end of Subsection 2.2 “Grain Growth and Strain Hardening Equations”. Accordingly, the introduction of Equation (11) (steady state condition) has been slightly modified.The referee is acknowledged for his(her) constructive comments.